# A protocol for the Hearing impairment in Adults: A Longitudinal Outcomes Study (HALOS)

Diana Tang[1¤]*, Yvonne Tran[1], Catherine McMahon[1], Jessica Turner[1], Janaki Amin[2], Kompal Sinha[3], Mohammad Nure Alam[3], Viviana Wuthrich[4], Kerry A. Sherman[4], Patrick Garcia[5], Rebecca Mitchell[5], Jeffrey Braithwaite[6], Greg Leigh[7], Shermin Lim[8], Giriraj Singh Shekhawat[8,9,10], Frances Rapport[6], Melanie Ferguson[11], Bamini Gopinath[1]

**1** Macquarie University Hearing, Department of Linguistics, Macquarie University, Sydney, NSW, Australia, **2** Department of Health Sciences, Macquarie University, Sydney, NSW, Australia, **3** Department of Economics, Macquarie University, Sydney, NSW, Australia, **4** School of Psychological Sciences, Macquarie University, Sydney, NSW, Australia, **5** Department of Management, Macquarie University, Sydney, NSW, Australia, **6** Australian Institute of Health Innovation, Macquarie University, Sydney, NSW, Australia, **7** Macquarie School of Education, Macquarie University, Sydney, NSW, Australia, **8** College of Nursing and Health Sciences, Flinders University, Adelaide, SA, Australia, **9** Ear Institute, University College London, London, United Kingdom, **10** Tinnitus Research Initiative, Regensburg, Germany, **11** Ear Science Institute Australia, Subiaco, WA, Australia

¤ Current address: Macquarie University, Sydney, NSW, Australia
* d.tang@mq.edu.au

**Data Availability Statement:** No datasets were generated or analysed during the current study. All relevant data from this study will be made available upon study completion.

## Abstract

### Background

Often considered an "invisible disability", hearing loss is one of the most prevalent chronic diseases and the third leading cause for years lived with disability worldwide. Hearing loss has substantial impacts on communication, psychological wellbeing, social connectedness, cognition, quality of life, and economic independence. The Hearing impairment in Adults: a Longitudinal Outcomes Study (HALOS) aims to evaluate the: (1) impacts of hearing devices (hearing aids and/or cochlear implants), (2) differences in timing of these interventions and in long-term outcomes between hearing aid and cochlear implant users, and (3) cost-effectiveness of early intervention for adult-onset hearing loss among hearing device users.

### Materials and methods

HALOS is a mixed-methods study collecting cross-sectional and longitudinal data on health and social outcomes from 908 hearing aid and/or cochlear implant users aged ≥40 years, recruited from hearing service providers across Australia. The quantitative component will involve an online survey at baseline (time of recruitment), 24-months, and 48-months and will collect audiological, health, psychosocial, functional and employment outcomes using validated instruments. The qualitative component will be conducted in a subset of participants at baseline and involve semi-structured interviews to understand the patient journey and perspectives on the Australian hearing service model.

**Funding:** BG, DM, CM, KS, JB, YT, HC, GL, FR, VW, JA, RM, PG, JB, MF, KS, DT - Cochlear Limited and Macquarie University https://www.cochlear.com/au/en/home https://www.mq.edu.au/ Cochlear Limited has not played or will play a role in the design, execution or reporting of the study. Through the authors, Macquarie University will have a role in study design, data collection and analysis, decision to publish and preparation of manuscripts.

**Competing interests:** The authors have declared that no competing interests exist.

## Ethics

This study has been approved by the Macquarie University Human Research Ethics Committee (ID: 11262) and Southern Adelaide Local Health Network (ID: LNR/22/SAC/88).

Dissemination of results: Study findings will be disseminated to participants via a one-page summary, and to the public through publications in peer-reviewed journals and presentations at conferences.

## Trial registration

Australia New Zealand Clinical Trial Registry (ANZCTR) registration number: ACTRN12622000752763.

## Introduction

Untreated hearing loss in midlife (45–65 years) was identified by a 2020 Lancet report as the biggest modifiable risk factor for a future dementia diagnosis [1]. As one of the most prevalent chronic diseases with one in five people affected, it is also the third leading cause for years lived with a disability worldwide [2, 3]. Despite its high prevalence, adult-onset hearing loss is largely under-recognised and often considered an "invisible disability". However, hearing loss substantially and negatively impacts communication, leading to detrimental effects on psychological health, social connectedness, and quality of life (QoL) for individuals, their families and society [4–6]. Hearing loss is also a significant economic burden in Australia with direct costs estimated to be $15.9 billion annually [7].

Current gaps in population-based surveillance efforts pertinent to adult-onset hearing loss include a lack of prospective data on the impact of hearing interventions such as cochlear implants (CI) and/or hearing aids (HA). Rather than focusing solely on auditory and speech outcomes, there is a critical need to adopt a public health approach that seeks to promote health through understanding how interpersonal and social functioning, QoL, cognitive health, mental wellbeing, independence, education and employment, are associated with the different hearing intervention pathways. Moreover, well-designed longitudinal studies that adequately control for confounders (e.g., sociodemographic indices and comorbidities) are imperative in establishing the impact and management of adult-onset hearing loss on individuals, families, and society. Such studies can help build evidence-based strategies to ultimately enable and support more holistic management of hearing loss at the population level.

These gaps are addressed by this internationally unique observational study—the Hearing impairment in Adults: a Longitudinal Outcomes Study (HALOS). This study aims to:

1. Evaluate the impacts of treating hearing loss on health (e.g., QoL, cognition, depression/mood, functional status), interpersonal relationships, education, and work;

2. Examine differences in long-term outcomes within and between groups of hearing device users;

3. Determine the impact from the timing of audiological intervention and the effectiveness of earlier intervention on outcomes;

4. Understand the patient journey through healthcare and identify the facilitators and barriers to accessing and using a hearing device; and

5. Estimate the cost-effectiveness of early intervention–through hearing aid(s) or cochlear implant(s) for hearing loss.

## Materials and methods

This study has been approved by the Macquarie University Human Research Ethics Committee (ID: 11262) on 12 May 2022 and the Southern Adelaide Local Health Network (ID: LNR/22/SAC/88) on 13 September 2022. The study has also been registered with the Australia New Zealand Clinical Trial Registry (ANZCTR) registration number: ACTRN12622000752763.

An Expert Advisory Group comprising of hearing health representatives from academia, service provider organisations, industry bodies and consumer groups has been established at the outset of this study. The Expert Advisory Group will guide study governance, design, data collection and analysis, and the dissemination of findings as well as monitor progress according to clear research principles. The Expert Advisory Group will develop a set of Terms of Reference at the outset of the study. HALOS is a mixed-methods study which will collect both cross-sectional and longitudinal data on health and social outcomes from hearing device users. The quantitative component will involve the administration of an online survey to the target population. Participants will be surveyed at three time-points, (1) baseline (time of recruitment), (2) 24-month follow-up, and (3) 48-month follow up. Audiological, health, psychosocial, and functional outcomes will be measured using validated instruments. The qualitative component will follow a first-come, first-serve, self-selected sampling approach and involve a subset of participants from the quantitative component at baseline. It will take the form of semi-structured interviews supported by an interview schedule.

## Recruitment

Participants will mainly be recruited from the following hearing service providers who have agreed to be involved in the project and to provide audiometric data for consenting participants. These providers are: NextSense, Hearing Australia, Audika, MQ Health Speech and Hearing Clinic, Amplifon, Bay Audio, Neurosensory, Flinders University (Flinders Health2Go), Southern ENT and Adelaide Sinus Centre, South Australia Speech and Hearing Centre, Speak Hear Speech and Hearing Services, the Royal Victorian Eye and Ear Hospital, and Brad Hutchinson Hearing. Providers will be given study flyers and participant information sheets on HALOS which can be displayed in clinics and provided to eligible clients during their visit. The study will also be advertised to potential volunteers through consumer support groups, existing University volunteer research databases and other University platforms such as websites and social media. The eligibility criteria include: 1) aged ≥40 years; 2) wears a hearing device in at least one ear; 3) sufficient English language competency to complete the online survey; and 4) able to give informed consent. Flyers and participant information sheets will include the contact details of the study team and the study website for interested individuals to express their interest in participating and/or seek further information. Hearing service providers will also have the option of sending out email invitations to members of their client database who have given consent to be contacted for research purposes.

Interested participants will be screened for eligibility via telephone interview or response to a checklist delivered electronically (SMS or email) and confirm their understanding of the commitments involved in participation. Those who meet the initial criteria and express continued interest in participating will proceed with the next steps of the respective study component/s. For the quantitative component, participants will be emailed a link to the consent form and online survey. For a subset of these participants who have also agreed to participate in the telephone-based qualitative component, a research team member will arrange a phone call to obtain audio-recorded verbal consent before commencing the interview.

## Quantitative component

Data from the quantitative component will be used to address the first three study aims, which are to evaluate the: 1. impacts of treating hearing loss on health (e.g., QoL, cognition, depression/mood, functional status), interpersonal relationships, education, and work; 2. differences in long-term outcomes within and between groups of hearing device users; 3. impact from the timing of audiological intervention and the effectiveness of earlier intervention on outcomes.

Participants will be sent a link to the quantitative survey and asked to read and sign the online consent form. Within the consent form, there is an option for participants to sign an 'Authorisation to Release Information' form that will permit the study team to collect retrospective audiometric data from the participant's hearing service provider. Once the consent form has been signed, participants will be redirected to a ~60-minute self-administered online survey developed in Research Electronic Data Capture (REDCap). Participants will have two weeks to complete the online survey and will be able to log in at any time during this period. A paper version will be provided (e.g., posted or emailed) to those who cannot complete the online version of the survey. A research team member will then manually enter these responses into REDCap. Upon completion of the survey, participants will be sent a $30 Coles-Myer gift voucher as compensation for their time commitment to this study.

## Audiometric data

The most recent audiometric data from the last two years will be obtained for all participants through their authorised consent to release this information from their hearing service provider. This will occur via the study team providing participating hearing service providers with an aggregated list of participant names who were sourced from their clinic(s). The collected data will include audiometric thresholds for air- and bone-conduction stimuli in both ears for octave frequencies at 0.25 to 8.0 kHz and/or four-frequency average hearing loss (4FAHL) data, measured at 0.5, 1, 2 and 4 kHz.

## Hearing and social outcome measures

We followed a biopsychosocial framework [8] when choosing the measures to be included in this survey. The biopsychosocial framework views health or illness as a product of biological characteristics, behavioural factors, and social conditions [8]. It is the interrelation of these components that leads to outcomes for health conditions. The survey items and validated questionnaires belonging to each of the biopsychosocial framework categories are shown in Fig 1. Broadly, the survey includes the following categories–demographic and hearing-related items; health-related quality of life; patient-reported outcomes; hearing-related scales; tinnitus; cognitive function; falls risk; frailty; employment-related scales; unmet needs; food insecurity and interpersonal relationship functioning. Details about the items and instruments used are shown in Table 1. Where appropriate, survey items have been modified for adult hearing device users (e.g., replacing 'hearing aid' with 'hearing device'). Responses to this survey will provide measurable population-based data on the impacts of hearing loss on relationships and other areas of life including education, employment, work, and career. These data will quantify the extent that hearing loss impacts individuals and society and help guide the allocation of resources to better support the needs of adults with hearing loss.

The online survey will primarily be hosted on REDCap which will have an embedded link to the self-administered Cogstate Brief Battery [17] hosted on an external website. The online survey has been tested by five adults aged 40+ with hearing loss to evaluate the suitability of the included questions, the online administration via REDCap and the time commitment

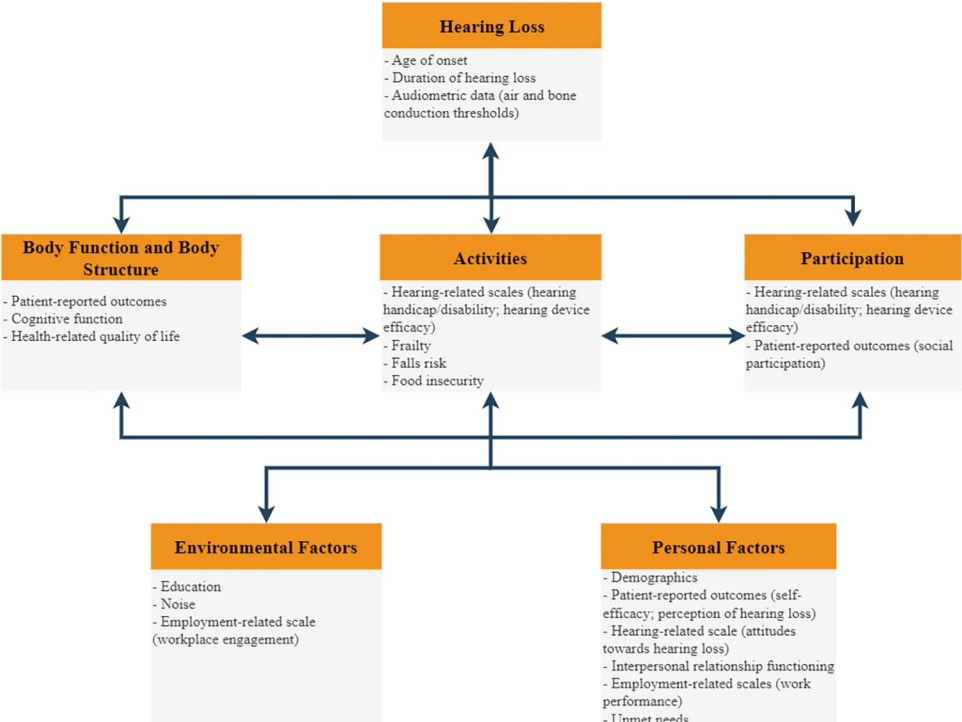

**Fig 1. Biopsychosocial model and the survey data belonging to each of the model categories.** This figure summarises the measures captured in HALOS relevant to the biopsychosocial model. The HALOS measures include demographic and hearing-related items; health-related quality of life; patient-reported outcomes; hearing-related scales; tinnitus; cognitive function; falls risk; frailty; and employment-related scales; unmet needs; and interpersonal relationship functioning.

involved. Based on their feedback, the online survey has been further refined ahead of administering to the study participants.

## Qualitative component

Data from the qualitative component will be used to address the fourth study aim, which is to better understand the healthcare journey for an adult with hearing loss, and identify the facilitators and barriers to accessing and using a hearing device.

At baseline, we will conduct one-to-one 30-minute interviews with a subset of participants from the quantitative component using a semi-structured interview guide (S1 Appendix). Participants will be enrolled on a first-come first-served basis until data saturation is achieved. Through these interviews participants will be asked to give a general overview of their experiences of the patient journey through healthcare and navigation of the hearing health service pathway. They will then be asked to broadly describe their preferences and concerns about the current hearing service model, and hearing interventions and devices provided. Participants will subsequently be asked for suggestions as to what aspects of the hearing health service pathway could be improved or modified. Open-ended questioning will allow participants to expand on their thoughts. Interviews will be audio-recorded, transcribed verbatim by an approved transcription service, and checked against the audio recording for accuracy.

## Sample size

Participants involved in the qualitative component will be recruited until data saturation in each group (i.e., CI versus HA) is achieved. According to a recent qualitative study involving

**Table 1. Outcome measures in HALOS.**

| Category | Items/Instruments |
|---|---|
| **Demographic and hearing-related items** | Demographic items (such as age, sex, education levels, marital status, and employment), contact details, comorbidities at baseline (such as, cardiovascular disease, cancer, diabetes, hypertension, cognitive decline), relevant information on hearing loss (such as age at onset of hearing loss, duration of hearing loss, age at implantation and/or initial use of hearing aid(s)), details about hearing services and/or devices received through the Australian Government Hearing Services Program, and details about access to consumer groups (such as Cicada, Deafness Forum, Deafness Foundation, and SoundFair). |
| **Health-related quality of life** | Health Utilities Index-3 (HUI-3) [9] |
| **Patient-reported outcome measures** | PROMIS™-43 Profile [10]<br>Social Participation Restrictions Questionnaire (SPaRQ) [11]<br>Brief Illness Perception Questionnaire (IPQ) [12] |
| **Hearing-related scales** | Hearing Handicap Inventory for Elderly–Screening version (HHIE-S) [13]<br>Speech, Spatial and Qualities of Hearing Scale (SSQ) [14]<br>International Outcome Inventory for Hearing Aids (IOI-HA) [15]<br>Attitudes towards Loss of Hearing Questionnaire (ALHQ) [16] |
| **Tinnitus** | Four questions to capture presence of tinnitus and its severity |
| **Cognitive function** | Cogstate Brief Battery consisting of a Detection Test, Identification Test, One Card Learning Test and One Back Test [17] |
| **Falls risk** | Two general questions to capture the presence of any falls in the past 12 months (yes/no), followed by a question to report the number of falls if the participant indicated a 'yes' response. |
| **Frailty** | FRAIL (Fatigue, Resistance, Ambulation, Illness and Loss of weight) scale [18] |
| **Employment-related scales** | Utrecht Work Engagement Scale (UWES-9) [19]<br>Individual Work Performance Questionnaire (IWPQ) [20]<br>Three items developed by Hellgren et al. (1998) related to quantitative job security [21] |
| **Unmet needs** | Short-Form Survivor Unmet Needs Survey (SF-SUNS) [22] |
| **Interpersonal relationship functioning** | Four-item relationship subscale developed by Donovan-Kicken and Caughlin [23] |
| **Food security survey** | 12 items which assess food insecurity [24] |

adults with a hearing loss, approximately 14 participants per group are likely to be required to reach data saturation [25]. For the quantitative component, we will use linear mixed-effects models as the primary analysis tool as it will take into consideration the longitudinal and dependent nature (repeated measures) of the data collected. Therefore, sample size calculations were based on a mixed model of repeated measures with a general correlation structure [26]. The sample size calculations were conducted using the 'longpower' R package [27]. The calculations assumed a group (CI versus HA) allocation ratio of 1:1, correlation between time points (rho) of 0.5, alpha of 0.05 and power of 0.8 [27]. Using participant drop-out estimations, attrition rates (loss to follow-up) were estimated as 10% in the first follow-up and a further 20% for the last follow-up [28]. Based upon available data [29], using health related QoL as the outcome measure and a conservative effect size (Cohen's d) of 0.3, a sample size of 454 per group (accounting for loss to follow-up) is needed to detect differences between the CI versus HA groups in the outcome of interest (i.e., QoL) over time.

## Statistical analyses

For the quantitative component, statistical analyses will be conducted using SPSS (Version 27 or later) and R (Version 4.3 or later). Descriptive statistics will be used to summarise the dataset and provide central tendency statistics. To understand the complex relationships between treating hearing loss and health and social outcomes (i.e., Aim #1), structural equation

modelling and path analyses will be conducted to examine direct and indirect pathways. To test for fixed effects for the two groups with consideration of the random effects from subjects (i.e., Aim #2), linear mixed effects regression models will be used. Differences within and between the two groups over time, as well as interaction effects between health and social outcomes will also be examined. Confounding factors e.g., age, sex, baseline pure-tone average thresholds, duration of hearing device use, will be included and adjusted for in the final models. To understand the patterns, trajectories and profiling developmental pathways for hearing impairment (i.e., Aim #3), group-based trajectory modelling (GBTM) will be used. An extension of finite mixture modeling techniques, GBTM maximises information available in multivariate longitudinal data to track the course of an outcome and assess the heterogeneity in the population allowing for more precise individual classifications into various groups that comprise of that taxonomy.

For the qualitative component (addressing aim #4), interview transcripts will be imported into NVivo software and analysed using inductive thematic analysis. Two research team members will independently undertake the following steps to identify the key themes from the data: (1) familiarisation with the data, (2) creation of codes, (3) identification of themes, (4) reviewing of themes, and (5) defining and naming of themes [30]. These steps form part of an iterative process and will be performed concurrently with the interviews to establish the point of data saturation.

## Economic analyses

The economic analysis will provide data to address the fifth study aim, which is to estimate the cost-effectiveness of early intervention through hearing aid(s) and/ or cochlear implant(s) for hearing loss. It will inform policymakers on the effectiveness and productivity of hearing devices to increase the efficacy of public and private resources employed in hearing intervention. HALOS data will also provide useful information required to evaluate the economic benefit and health inequity implications of hearing devices for adult-onset hearing loss. Adopting a societal perspective [31] with a lifetime horizon, HALOS will undertake an economic analysis to examine the effects of hearing device usage on QoL (using the Health Utility Index-3 (HUI-3 [9]). To quantify the changes in QoL following hearing device(s) uptake, a summary score will be assessed from the HUI-3 questionnaire [9].

A cost-benefit analysis (CBA) will be conducted to assess the efficiency of hearing devices for hearing loss. There could be heterogeneity in the patients' experience with CI and HA due to differences in direct costs, resource utilisations, utility, and sociodemographic backgrounds. HALOS aims to collect clinical data on different types of CI, which will allow for a comparison of the outcomes. This project will conduct a within-group (CI and HA) CBA based on the types of hearing device (unilateral versus simultaneous bilateral CI) and hearing severity (mild, moderate or severe hearing loss for HA) to see the differences in benefit-cost ratios. The economic analyses of hearing device usage will provide evidence regarding the effectiveness of early interventions and the variations in benefit-cost ratios by the CI type (e.g., unilateral and sequential bilateral).

## Follow-up

Using the provided contact details obtained at baseline, participants who consented to being followed up at baseline will be reapproached to complete their 24- and 48-month follow-up. Participants will be emailed a new link to access the respective follow-up survey that will include the same health and social outcome measures assessed at baseline. These longitudinal data will be linked to the participants' baseline study ID number to allow capture data across

these timepoints, which will help contribute to achieving the first three study aims that were previously mentioned.

## Discussion

HALOS will fill critical gaps in knowledge on the temporal impacts of using cochlear implants and/or hearing aids in adults. Several key outcomes of the study are expected. These include advancing the evidence-base on the effects of hearing loss and hearing devices, and in turn contributing towards shaping evidence-based clinical guidelines for best practice management of adults with hearing loss; informing the design of more sensitive tools that can be used broadly to measure the benefits of cochlear implants and hearing aids; and providing transparent and comparable, evidence-based information to policymakers in support of guiding informed decisions on the provision of devices for treating hearing loss.

The overall risk of this study is low; however, participants may experience some minor tiredness from completing the survey. To minimise this risk, participants will be instructed that they can complete the survey over multiple sessions within the two-week period to minimise any potential tiredness. Participants will be able to stop at any time and withdraw from the study without giving a reason. Some participants may also experience feelings of distress as a result of completing the questionnaire. Participants will be advised to contact the research team, which includes an Accredited Mental Health First Aider and clinical psychologists, for advice if they experience distress. A list of organisations who can provide support will also be provided to participants.

The results of the study will be disseminated by a one-page summary of the study findings to participants via post or email for those who opt in to receive this on their consent form. Study findings will also be disseminated through publications in peer-reviewed journals, consumer group newsletters, and affiliated and partner institution networks, as well as through presentations at conferences, in order to reach a wide range of relevant stakeholders and consumers.

## Supporting information

**S1 Appendix. Semi-structured interview guide.**
(DOCX)

## Acknowledgments

The authors acknowledge the support of the following hearing service providers for this study: Cochlear Ltd, NextSense, Hearing Australia, National Acoustics Laboratories, Audika, MQ Health Speech and Hearing Clinic, Amplifon, Bay Audio, Neurosensory, Flinders University (Flinders Health2Go), Southern ENT and Adelaide Sinus Centre, South Australia Speech and Hearing Centre, Speak Hear Speech and Hearing Services, the Royal Victorian Eye and Ear Hospital and Brad Hutchison Hearing.

## Author Contributions

**Conceptualization:** Diana Tang, Yvonne Tran, Catherine McMahon, Janaki Amin, Kompal Sinha, Viviana Wuthrich, Kerry A. Sherman, Patrick Garcia, Rebecca Mitchell, Jeffrey Braithwaite, Greg Leigh, Frances Rapport, Melanie Ferguson, Bamini Gopinath.

**Funding acquisition:** Diana Tang, Yvonne Tran, Catherine McMahon, Janaki Amin, Kompal Sinha, Viviana Wuthrich, Kerry A. Sherman, Patrick Garcia, Rebecca Mitchell, Jeffrey Braithwaite, Greg Leigh, Frances Rapport, Melanie Ferguson, Bamini Gopinath.

**Methodology:** Diana Tang, Yvonne Tran, Catherine McMahon, Janaki Amin, Kompal Sinha, Mohammad Nure Alam, Viviana Wuthrich, Kerry A. Sherman, Patrick Garcia, Rebecca Mitchell, Jeffrey Braithwaite, Greg Leigh, Shermin Lim, Giriraj Singh Shekhawat, Frances Rapport, Melanie Ferguson, Bamini Gopinath.

**Writing – original draft:** Diana Tang, Yvonne Tran.

**Writing – review & editing:** Catherine McMahon, Jessica Turner, Janaki Amin, Kompal Sinha, Mohammad Nure Alam, Viviana Wuthrich, Kerry A. Sherman, Patrick Garcia, Rebecca Mitchell, Jeffrey Braithwaite, Greg Leigh, Shermin Lim, Giriraj Singh Shekhawat, Frances Rapport, Melanie Ferguson, Bamini Gopinath.

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
