## [Decision Letter · Decision Letter 0]

7 Feb 2023

PONE-D-22-30154A protocol for the Hearing impairment in Adults: a Longitudinal Outcomes Study (HALOS)PLOS ONE

Dear Dr. Tang,

Thank you for submitting your manuscript to PLOS ONE. After careful consideration, we feel that it has merit but does not fully meet PLOS ONE’s publication criteria as it currently stands. Therefore, we invite you to submit a revised version of the manuscript that addresses the points raised during the review process.

We look forward to receiving your revised manuscript.

Kind regards,

Ari Samaranayaka, PhD

Academic Editor

PLOS ONE

Journal Requirements:

Additional Editor Comments (if provided):

1). Can authors please define the “baseline” for readers? I assume it is the time of recruitment, not the time of first using the hearing devices. Then period of hearing devices use at the baseline is vary between participants. How authors hope to manage that? This is applicable because concept of longitudinal design is based on outcomes likely to vary with time.

2). Cover letter says this is a pilot study, can I assume that is a typo because the manuscript is for a full study?

Reviewers' comments:

Reviewer's Responses to Questions

**Comments to the Author**

1. Does the manuscript provide a valid rationale for the proposed study, with clearly identified and justified research questions?

Reviewer #1: Partly

2. Is the protocol technically sound and planned in a manner that will lead to a meaningful outcome and allow testing the stated hypotheses?

Reviewer #1: Partly

3. Is the methodology feasible and described in sufficient detail to allow the work to be replicable?

Reviewer #1: No

4. Have the authors described where all data underlying the findings will be made available when the study is complete?

Reviewer #1: Yes

5. Is the manuscript presented in an intelligible fashion and written in standard English?

Reviewer #1: Yes

6. Review Comments to the Author

You may also provide optional suggestions and comments to authors that they might find helpful in planning their study.

Reviewer #1: This is a description of a study protocol regarding hearing impairments in adults based on HALOS. The paper does not contain any results – not either preliminary results – but I think the ideas behind this upcoming study could be of interest for other researchers. My main comment is the structure of the protocol. I would have preferred that the paper was structured based on the 5 overall aims. So, for each of the 5 aims please describe the study population and the statistical methods used to investigate that specific question. It would help the reader to understand what you expect to do, and to better understand each of the specific aims – for instance aim 3 is unclear to me. Moreover, a language check would be preferable before publication (see for instance the fourth line of the introduction).

7. PLOS authors have the option to publish the peer review history of their article (what does this mean?). If published, this will include your full peer review and any attached files.

Reviewer #1: No

---

## [Author Response · Author response to Decision Letter 0]

22 Feb 2023

Response to reviewers has been uploaded in the document 'Response to reviewers'.

---

## [Editor Report · Decision Letter 1]

3 Mar 2023

A protocol for the Hearing impairment in Adults: a Longitudinal Outcomes Study (HALOS)

PONE-D-22-30154R1

Dear Dr. Tang,

We’re pleased to inform you that your manuscript has been judged scientifically suitable for publication and will be formally accepted for publication once it meets all outstanding technical requirements.

Kind regards,

Ari Samaranayaka, PhD

Academic Editor

PLOS ONE
---

## [Editor Report · Acceptance letter]

6 Mar 2023

PONE-D-22-30154R1 

A protocol for the Hearing impairment in Adults: a Longitudinal Outcomes Study (HALOS) 

Dear Dr. Tang:

I'm pleased to inform you that your manuscript has been deemed suitable for publication in PLOS ONE. Congratulations! Your manuscript is now with our production department. 

Kind regards, 

on behalf of

Dr. Ari Samaranayaka 

Academic Editor

PLOS ONE